# Robust Free-Space Optical Communication Utilizing Polarization for the Advancement of Quantum Communication

**DOI:** 10.3390/e26040309

**Published:** 2024-03-30

**Authors:** Nicholas Savino, Jacob Leamer, Ravi Saripalli, Wenlei Zhang, Denys Bondar, Ryan Glasser

**Affiliations:** 1Department of Physics and Engineering Physics, Tulane University, New Orleans, LA 70118, USA; nsavino@tulane.edu (N.S.); jleamer@tulane.edu (J.L.); zhangw47@corning.com (W.Z.); 2Directed Energy Research Center, Technology Innovation Institute, Abu Dhabi P.O. Box 9639, United Arab Emirates; ravikiran.saripalli@tii.ae

**Keywords:** polarization, DOP, optics, turbulence, FSO, simulation, SOP, COMSOL

## Abstract

Free-space optical (FSO) communication can be subject to various types of distortion and loss as the signal propagates through non-uniform media. In experiment and simulation, we demonstrate that the state of polarization and degree of polarization of light passed though underwater bubbles, causing turbulence, is preserved. Our experimental setup serves as an efficient, low cost alternative approach to long distance atmospheric or underwater testing. We compare our experimental results with those of simulations, in which we model underwater bubbles, and separately, atmospheric turbulence. Our findings suggest potential improvements in polarization based FSO communication schemes.

## 1. Introduction

Free-space optical (FSO) communication is a widely used method of sending high data rate signals over long distances. It allows for wide bandwidths, licence free spectra, and high bit rates [1,2,3,4,5]. Atmospheric turbulence, however, poses a significant threat to FSO communication [1]. Signal attenuation resulting from turbulent fluctuations [6,7,8,9] can result in increased bit-error rates and data security concerns. Work is constantly being done to improve FSO communications such that it can become resistant to turbulence [10], for which many techniques require cumbersome active optics, advanced algorithms, and/or machine learning [11,12,13,14,15].

It has been shown that polarization can be used for atmospheric FSO [16]. The use of optical polarization in communication opens up the realm of quantum communication and quantum internet [17]. However, turbulence in the atmosphere can potentially produce impurities in quantum states. Understanding how turbulence affects polarization lays the groundwork for better understanding how turbulence may affect quantum states that are realized via optical polarization states.

Turbulence in the atmosphere due to the temperature differences between packets of air results in random fluctuations of the index of refraction *n*. These variations can be modeled as [18].
(1)$$n\left(\vec{r}\right)=n_{0}+n_{1}\left(\vec{r}\right),$$
where $\vec{r}$ denotes the position, $n_{0}\approx 1$ is the mean value of the refractive index, and $n_{1}\left(\vec{r}\right)$ are fluctuations. For atmospheric turbulence, $n_{1}\left(\vec{r}\right)$ is typically several orders of magnitude smaller than $n_{0}$ [19]. The effects that these fluctuations have on light passing through the atmosphere can be studied by considering the wave equation for the electric field $\vec{E}\left(\vec{r},t\right)$. For convenience, we consider monochromatic light with frequency ω and time dependence $e^{i\omega t}$ propagating through a source-free region of atmosphere. Then, Maxwell’s equation for the electric fields are reduced to [18].
(2)$$\nabla^{2}\vec{E}\left(\vec{r}\right)+\frac{\omega^{2}n{\left(\vec{r}\right)}^{2}}{c^{2}}\vec{E}\left(\vec{r}\right)+2\nabla \left(\vec{E}\left(\vec{r}\right)\cdot \nabla \ln\left[n\left(\vec{r}\right)\right]\right)=0,$$
where *c* is the speed of light in vacuum. The third term in Equation (Equation 2) couples the different components of the electric field, which weakens the polarization of the field. Because $n_{1}\left(\vec{r}\right)\ll n_{0}$, however, this depolarizing term tends to be very small and can often be ignored for visible light passing through the atmosphere [20]. We also confirm this via simulations in Appendix A. This would suggest that the polarization of light could serve as a useful degree of freedom in FSO communications, particularly given that polarization transformers have been shown to be able to manipulate the state and degree of polarization (SOP and DOP, respectively) [21] in ways that may be suitable for FSO communications [22]. It has also been shown that polarization can be used to encode and transmit information securely [23], and that states with a DOP of 1 are preserved in the presence of turbulence [16,24]. Utilizing these properties with methods like coherent binary polarization shift keying [25] may lead to improvements in FSO communications [26].

However, it can be difficult to develop these methods, as the experiments generally require the implementation of long distance atmospheric testing. An alternative approach could be the use of air bubbles in water, which are known to create intensity fluctuations that are often well-described by the log-normal distribution commonly used in studies of atmospheric turbulence [27,28,29,30]. Directly applying the atmospheric turbulence models to bubbles in water can be problematic as it overlooks key differences between the environments [31]. In particular, the refractive index model will be very different for a region of water with bubbles. This is because the change in refractive index from water to air (n=1.33 to n=1) is both much larger than the fluctuations in atmospheric turbulence and discontinuous rather than smoothly varying. While the mechanics of the underwater bubbles and atmospheric turbulence are different, the resulting intensity distributions of transmitted light appear to be similar. By passing light through underwater turbulence, we are not attempting to recreate exact conditions found in the atmosphere, but rather create conditions in which the index of refraction changes even more rapidly than in turbulent air. As such, the assumption that the depolarizing term in Equation (Equation 2) is negligible is not valid.

This point can be illustrated by considering a set of light rays passing through a region of water with air bubbles. When a ray collides with a bubble, some portion of it will be reflected while the rest will be refracted according to Snell’s law. Both reflection and refraction are known to change the SOP and DOP of light depending on the initial polarization and incident angle. For example, the reflected portion of unpolarized light hitting a bubble at the Brewster angle (∼$37^{\circ }$ for the water to air interface) will be completely polarized perpendicular to the plane of incidence while the refracted portion becomes slightly polarized. It’s clear that, in general, the SOP and DOP of a ray that encounters a bubble will not be preserved and thus the depolarizing term in Equation (Equation 2) is significant. However, it is possible that not all of the rays will hit a bubble. Thus, it may be possible that the average SOP and DOP of the rays remains well-preserved even though the polarization of individual rays is not.

In this article, we demonstrate experimentally and theoretically that the polarization of light is still well-preserved after passing through turbulent media. Section 2.1 describes the details of our experimental setup, along with out polarization tomography technique. The use of the underwater air bubble setups may speed up the design-test cycle for FSO communication methods. In Section 2.2 we describe our simulations. We also propose the use of DOP as a viable degree of freedom for FSO communication given that we find it is well preserved for even weakly polarized states. This may lead to improvements for polarization based modulation schemes and methods for communication.

## 2. Materials and Methods

### 2.1. Experiment

The experimental setup is shown in Figure 1. The desired partially polarized states are generated by a Mach-Zehnder interferometer, where a polarizing beamsplitter generates beams of horizontal and vertical polarizations, which are then recombined on a 50:50 non-polarizing beamsplitter (BS), such that they are separated by a small distance and not coaxially superposing. This allows for control over the DOP by adjusting relative intensity of each arm of the interferometer [21,32]. This is done by attenuating one arm by tuning a variable neutral-density filter. To change the basis from horizontal/vertical to an arbitrary basis, the light is passed through a quarter-wave plate (QWP), half-wave plate (HWP), and another QWP.

The light passes through approximately 1 m of fiber prior to entering free-space. The preparation of states requires approximately 1 m of propagation through free-space. Once in the desired polarization state, the light is transmitted through free-space for approximately 3 m. When measuring the effects of turbulence, the light passes through the entire length of the water tank, which is 30 cm long and holds approximately 10 L of water when full (the tank measures approximately 30 cm by 15 cm by 20 cm and is made out of uncoated glass). Because the propagation distance is on the order of meters, we assume any losses related to other link parameters will be negligible.

We generate underwater turbulence tank with submerged bubblers. As mentioned previously, underwater bubbles are known to cause fluctuations in the intensity of propagating light [27,28,29,30]. While we choose to use bubbles as a means of generating turbulence, which produces, changes in indices of refraction when the light passes through the surface interfaces, there has also been significant research in turbulence generated with fog chambers [33,34], which create continuously varying indices of refraction. These fluctuations can be characterized using the scintillation index [9].
(3)$$\begin{array}{c} \sigma_{I}^{2}=\frac{\langle I^{2}\rangle -{\langle I\rangle }^{2}}{{\langle I\rangle }^{2}}, \end{array}$$
where *I* is the experimentally measured intensity and 〈·〉 denotes the temporal average. The photodiode detector used is a ‘bucket detector’; it only measures the intensity. No wavefront measurements are conducted. The scintillation index for light passing through underwater bubbles varies depending on the size and number of bubbles, but typical values are between 0.1 and 1.0 [28,29]. Over the course of the experiment and including all intensity projection measurements, we observe an average $\sigma_{i}^{2}=0.5\pm 0.1$.

To calculate the resulting power at the detector, $P_{rec}$, we can summarize the free-space link budget as follows
(4)$$P_{rec}=P_{trans}+G_{trans}-L_{trans}-L_{fs}-L_{turb}+G_{rec}-G_{rec}.$$ The approximate link parameters are expressed in the Table 1.

The transmitter loss comes from the fiber optic cable and state preparation. The loss from the fiber is uniform across all measurements and the values are not expected to have any significant impact on the data. The variability in the transmission loss comes from the VND filter, which alters the SOP. While there may be very small loss associated with the free-space propagation, it is negligible in comparison to the loss due to the turbulence and electronic noise of our detectors. Loss due to turbulence varies slightly between trials. Loss at the receiver is variable due to the nature of our setup; depending on the SOP and the orientation of the QWP and LP, we measure different intensities as expected. As we measure the normalized Stokes parameters in this experiment, the overall intensity and any global loss should not have an impact on the results.

Tomography measurements on the polarization matrix can be conducted using a polarimeter to obtain the Stokes parameters of our beam. However, we use an intensity detection scheme, illustrated in Figure 1, that only requires a QWP, linear polarizer, and photodetector to obtain polarization projection measurements [35,36]. Four intensity measurements are performed: $I(0^{\circ },\;0^{\circ })$, $I(0^{\circ },\;90^{\circ })$, $I(0^{\circ },\;45^{\circ })$, and $I(45^{\circ },\;45^{\circ })$. I(ψ,ϕ) is the intensity measured by the photodetector with the fast axis of the QWP (in the detection scheme) at angle ψ, and the axis of transmission of the linear polarizer at angle ϕ (both with respect to the horizontal axis). With these intensity measurements, we can calculate the Stokes parameters [36]:(5)$$\begin{array}{c} s_{0}=I\left(0^{\circ },0^{\circ }\right)+I\left(0^{\circ },90^{\circ }\right),\\ s_{1}=I\left(0^{\circ },0^{\circ }\right)-I\left(0^{\circ },90^{\circ }\right),\\ s_{2}=2I\left(45^{\circ },45^{\circ }\right)-s_{0},\\ s_{3}=2I\left(0^{\circ },45^{\circ }\right)-s_{0}. \end{array}$$ Then from the Stokes parameters, we obtain the DOP of the states using
(6)$$\mathrm{DOP}=\frac{\sqrt{s_{1}^{2}+s_{2}^{2}+s_{3}^{2}}}{s_{0}}.$$

We take 120,000 intensity measurements over 24 s for each polarization projection, then take a time average of all measurements to obtain an average intensity to be used with Equation (Equation 6). It is important to recognize that our results rely on many measurements being recorded, then time averaged to show DOP is preserved. The setup in Figure 1 only allows us to take the intensity measurements independently at different times. We take all input measurements in which we remove the tank from the setup, then reproduce all states when the tank is placed in the path of the light in order to take output measurements. This is done to keep the nature of the underwater turbulence as consistent as possible when measuring different states. Due to the irreproducibility and chaotic nature of bubbles, enough data points to recover the entire intensity distribution must be taken. More details regarding the intensity distribution of the experimental results are presented in Appendix B.

### 2.2. Simulation

We use the COMSOL Ray Optics [37] package to simulate the experimental setup shown in Figure 1. The full code used can be found on Github [38]. In these calculations, we consider light propagating in the *x*-direction through a region of water that measures 100×100 cm. Because the speed of the bubbles in the experiment is negligible compared to the speed of light, we ignore the bubbles’ motion. We perform simulations for light that is initially vertically polarized and for light with an arbitrary initial state of polarization. In both cases, we consider five initial values for the DOP and run 100 simulations for each DOP. In every simulation, we consider 300 rays that are evenly spaced from x=35 cm to x=65 cm. To ensure that none of the rays can make it through the region of water without hitting bubbles, we create two layers of bubbles spanning the height of domain. The radius of each bubble is determined by sampling a uniform distribution between 0.1 and 1 cm. The *x* positions of the bubbles in the first layer are chosen by sampling a truncated normal distribution between 10 and 49 cm, while *x* positions of the bubbles in the second layer are selected from a truncated normal distribution between 51 and 90 cm. The gap between the layers is chosen to avoid issues with overlapping the bubbles in COMSOL. In the simulations, we consider only the rays that reach x=100 cm to be detected to mimic the experimental setup. The COMSOL Ray Optics package keeps track of the Stokes parameters of each ray throughout the simulation, so calculating the initial and final DOP for a given simulation can be done by calculating the average $s_{1}$, $s_{2}$, $s_{3}$, and $s_{0}$ of the detected rays at the beginning and end of the simulation and using Equation (Equation 6). In Appendix B it is shown that the intensity distributions from the simulations agree with the experimental measurements.

As mentioned in Section 1, we also used the COMSOL Ray Optics software to model light passing through atmospheric turbulence. The details of these calculations and the corresponding results can be found in Appendix A.

## 3. Results

We obtain experimental results for five input states that are linearly polarized with different DOPs. In Figure 2, we plot the values of the input and output DOP. The diagonal black line in Figure 2 represents equal input and output DOP. The experimental results are compared with the results from the simulation with stationary bubbles. The individual experimental Stokes parameters are shown in Table 2. We note that there is significant intensity attenuation of the beam as it propagates through the bubbles and water, which is to be expected. We see a reduction in average intensity by a factor of about 20, while the DOP is well-maintained. In Figure 3, we show the location of the input and output states in the Poincare sphere representation of normalized in three-dimensional Stokes space. We see the corresponding input and output states lie very close to each other on the Poincare sphere.

While we measure all Stokes parameters, a full tomography of the polarization matrix is not necessarily required for communication purposes, given appropriate encoding and decoding protocols are chosen. If we use the input states shown in Figure 3, only measurement along the $s_{1}$ Stokes parameter is necessary to determine the encoded value. Thus, the measurement can be done in a more robust manner, with only two intensity measurements, $I(0^{\circ },\;0^{\circ })$ and $I(0^{\circ },\;90^{\circ })$. Due to our choice of basis, where $s_{2}$ and $s_{3}$ are approximately 0, we can recover the DOP with only $s_{0}$ and $s_{1}$. This measurement can be generalized to all directions on the Poincare sphere by exploiting its symmetry, as there is no preferred direction on the Poincare sphere.

We can also choose some arbitrary basis to generate arbitrary states. The individual Stokes parameters, when measuring states along an arbitrary basis are shown in Table 3. Thus, our results suggest that SOP is also preserved. The input and output DOPs, along with the locations of each state on the Poincare sphere are shown in Figure 4.

## 4. Discussion

We show that SOP and DOP are preserved when passed through experimentally generated underwater turbulence, simulated underwater bubbles, and simulated atmospheric turbulence. In experiment, we see that minor unexpected changes to the properties of the bubbles between trials do not significantly affect the results. We also note that the DOP is not exactly preserved, as the input states are slightly outside of confidence interval for most measured output states. We suspect this is caused by error in experimental state generation and measurement, and is not a direct result of the bubbles. A more sensitive state generation and detection setup can reduce this error.

We can significantly decrease the time scales needed to obtain results if we collect multiple intensity measurements at the same time. Measurements can be made anywhere on the entire Poincare sphere if $I(0^{\circ },\;0^{\circ })$, $I(0^{\circ },\;90^{\circ })$, $I(0^{\circ },\;45^{\circ })$, and $I(45^{\circ },\;45^{\circ })$ are measured simultaneously. A polarimeter that uses a rotating wave-plate/polarizer scheme, where the intensity is continuously measured as a function of time as the wave-plate/polarizer spin can result in near instantaneous detection. The measurements would need to be taken at a faster rate than the fluctuations. Some time averaging may still be necessary unless the entire state observes uniform turbulence across the entire beam profile.

DOP and SOP show promise as a method of encoding information in FSO communication that is preserved when passed through atmospheric turbulence or underwater bubbles. The sender and receiver only need passive optics to encode and decode the respective data; no active optics are necessary. Turbulence resilience without the need of active optics could be particularly beneficial for advancements in ground-to-satellite communication, as bypassing active optics can result in significant weight savings [39]. We believe the methods presented here can work in tandem with other correction and security measures to continue to improve upon FSO communication. Due to the nature of our experiment, we believe these methods may also hold promise in improving underwater optical communication [40].

As discussed in Section 2.1, to encode onto an arbitrary polarization state, the light can be passed through a QWP, followed by a HWP, and then another QWP. Rotating these three components allows for any point within the Poincare sphere. Our results have potential to improve already existing polarization dependant communication schemes [22,41,42,43]. Encryption schemes similar to quantum key distribution [41] can also be performed, where the sender and receiver share a key which determines which bases to measure. Since DOP is a projection onto the three-dimensional Poincare sphere, it can potentially be used as a turbulence resistant communication mode, in which information is carried in both the base and value. As the Poincare sphere is analogous to the Bloch sphere, applications can potentially be implemented in the field of quantum information. Polarization multiplexing for space-to-ground optical communication has been studied [44].

Furthermore, we show the intensity fluctuations generated with our experimental setup are similar to models used to describe true atmospheric turbulence on the macro scale [9]. Results obtained in experiment and simulation agree with one another. Simulation results also suggest that our experimental method of generating bubbles will produce the same results as with continuously varying index of reflection media when examining the input and output states of polarization. Ongoing work will need to be done to verify that changing turbulence properties do not alter the DOP and SOP.

## Figures and Tables

**Figure 1 entropy-26-00309-f001:**
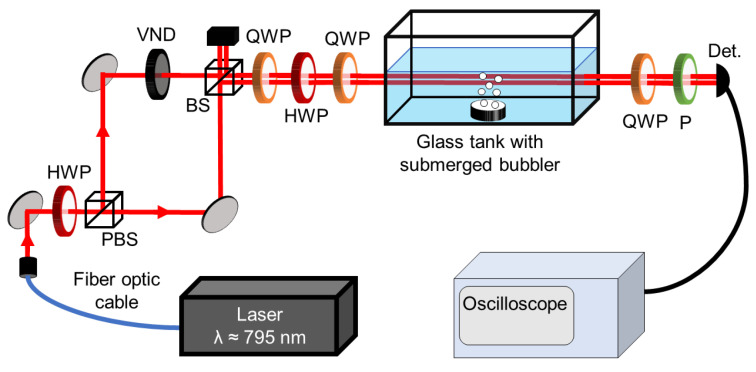
Experimental setup used to generate different polarization states, pass them through experimentally generated turbulence, and perform tomography measurements. Abbreviations: PBS = polarizing beamsplitter, QWP = quarter-wave plate, HWP = half-wave plate, P = linear polarizer, VND = variable neutral density filter, BS = 50:50 non-polarizing beamsplitter, Det. = photodetector.

**Figure 2 entropy-26-00309-f002:**
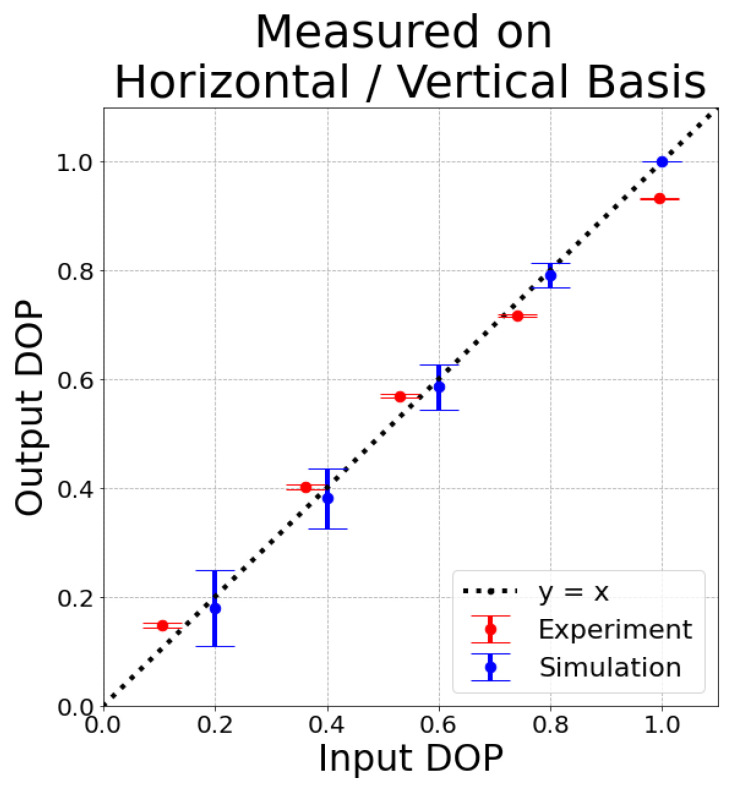
Measured DOP of the output states through underwater bubbles compared to the measured DOP of the input states for different partially polarized states. Input and output DOP are equal at the black like (y=x). Experimentally obtained data is shown in red and simulation results are shown in blue. The error bars represent 99% confidence interval.

**Figure 3 entropy-26-00309-f003:**
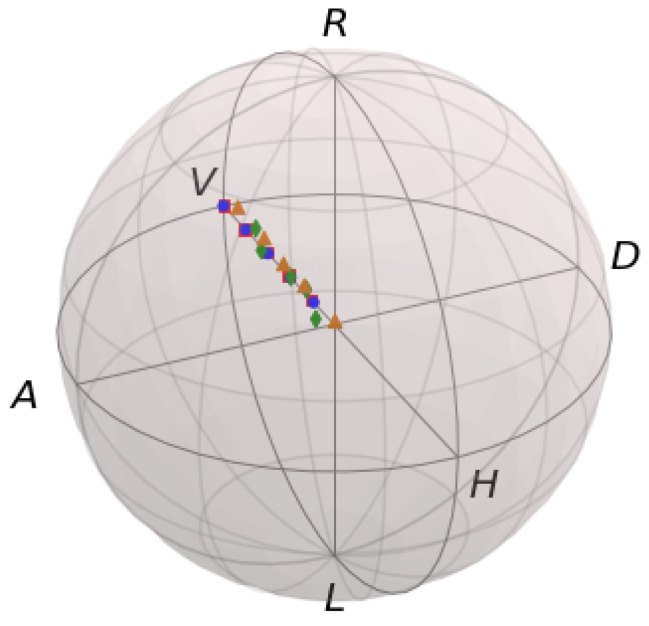
Input and output states in the Poincare sphere representation. Experimental input states are orange triangles, experimental output states are green diamonds, stationary bubbles simulation input states are red squares, stationary bubbles simulation output states are blue circles. On the Poincare sphere: D = diagonal, A = anti-diagonal, L = left, R = right, V = vertical, H = horizontal.

**Figure 4 entropy-26-00309-f004:**
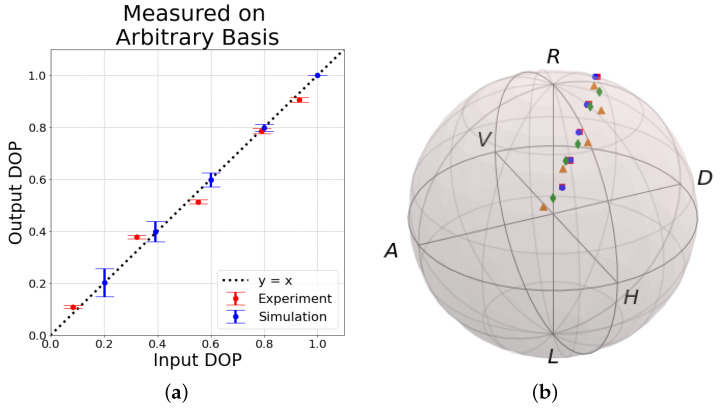
Measured DOP of the output states through underwater turbulence compared to the measured DOP of the input states for different arbitrary partially polarized states. Experimental and stationary bubbles simulation results are compared. (**a**) The output DOP with respect to the input DOP for an arbitrary basis. Input and output DOP are equal at the black like (y=x). Experimentally obtained data is shown in red and stationary bubbles simulation results are shown in blue. The error bars represent 99% confidence interval. We see DOP is preserved for arbitrary states. (**b**) Arbitrary input and output states in the Poincare sphere representation. Experimental input states are orange triangles, experimental output states are green diamonds, stationary bubbles simulation input states are red squares, stationary bubbles simulation output states are blue circles. On the Poincare sphere: D = diagonal, A = anti-diagonal, L = left, R = right, V = vertical, H = horizontal.

**Table 1 entropy-26-00309-t001:** Link parameters for the experimental setup. All values are approximate.

Parameter	Symbol	Value
Power transmitted	$P_{trans}$	≈24 dBm
Transmitter gain	$G_{trans}$	≈0 dB
Transmitter loss	$L_{trans}$	≈15 to 18 dB
Free-space loss	$L_{fs}$	≈0 dB
Turbulence related loss	$L_{turb}$	≈13 dB
Receiver gain	$G_{rec}$	≈0 dB
Receiver loss	$G_{rec}$	≈0 to 20 dB

**Table 2 entropy-26-00309-t002:** Measured mean normalized Stokes parameters of the input and output partially polarized states on the horizontal/vertical axis through experimental underwater turbulence.

State	$s_{1}/s_{0}$		$s_{2}/s_{0}$		$s_{3}/s_{0}$		DOP	
	Input	Output	Input	Output	Input	Output	Input	Output
0	0.09	0.14	−0.04	0.01	−0.04	−0.05	0.11	0.15
1	0.36	0.39	−0.05	−0.07	−0.03	−0.07	0.36	0.40
2	0.53	0.55	−0.04	−0.08	−0.02	−0.10	0.53	0.57
3	0.74	0.71	−0.05	−0.03	−0.02	−0.06	0.74	0.72
4	0.99	0.92	−0.06	−0.10	−0.02	−0.08	0.99	0.93

**Table 3 entropy-26-00309-t003:** Measured mean normalized Stokes parameters of the input and output arbitrary partially polarized states through experimental underwater turbulence.

State	$s_{1}/s_{0}$		$s_{2}/s_{0}$		$s_{3}/s_{0}$		DOP	
	Input	Output	Input	Output	Input	Output	Input	Output
0	0.03	0.04	0.06	−0.02	0.05	0.10	0.08	0.11
1	0.08	0.11	−0.11	−0.15	0.29	0.33	0.32	0.38
2	0.12	0.13	−0.32	−0.25	0.43	0.43	0.55	0.51
3	0.18	0.17	−0.45	−0.36	0.62	0.68	0.79	0.79
4	0.25	0.25	−0.43	−0.47	0.78	0.73	0.93	0.91

## Data Availability

The raw data supporting the conclusions of this article will be made available by the authors on request.

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
