# Peer review of "Robust Free-Space Optical Communication Utilizing Polarization for the Advancement of Quantum Communication"

_entropy, 2024, doi:10.3390/e26040309_

Round 1

Reviewer 1 Report

Comments and Suggestions for Authors

The article describes a protocol to verify the survival of polarisation states when passing through a turbulent environment. The system is explored both numerically and experimentally and the results are in reasonable agreement.

The paper is clear and easy to follow and scientifically sound for the testing part.

What is missing in the paper is a discussion of the topic announced by the title: there is no elaboration on the actual feasibility of a communication protocol implemented with their system.

In particular, the few sentences included in the conclusions are too general and obscure.

Before accepting the article for Entropy, the authors should make it more in keeping with the journal's theme. Otherwise, I suggest they change the title and move on to another journal.

Reviewer 2 Report

Comments and Suggestions for Authors

Authors present interesting experimental setup with polarization measurements and simulation results. My main comments are as follows since contribution is not very clear and the manuscript seems to have small impact:

-          Please emphasize on the main contribution. Is it the experimental setup? While it is alternative of the FSO links, there are similar works with gas chambers emulating atmospheric turbulence (Northumbria University worked on that, latest work was also shown by Technical University of Eindhoven, Chinese teams also had similar work with gas chamber);

-          I do not understand the discussion in the paragraph on lines 203-212. We use adaptive optics for completely different reasons, namely, to correct the wavefront in the receiving aperture, which is important especially for fiber coupling and coherent communications. I don’t understand the connection to maintained polarization. FEC is a different topic due to signal fades which also exist regardless of polarization.

-          Polarization multiplexing is already present in all 100Gbps class SFPs for fiber network. Actual experiment using dual polarization was demonstrated with TBIRD from MIT LL recently. Negligible atmospheric turbulence effects on DOP have been shown during SOTA experiments (http://www.nature.com/nphoton/journal/vaop/ncurrent/full/nphoton.2017.107.html), SOP can only be assumed not to change due to the LEO orbit movement.

Round 2

Reviewer 1 Report

Comments and Suggestions for Authors

The authors modified the text according to my suggestions and I believe the manuscript can be now accepted for publication.